# CD30 Expression and Its Functions during the Disease Progression of Adult T-Cell Leukemia/Lymphoma

**DOI:** 10.3390/ijms24108731

**Published:** 2023-05-13

**Authors:** Makoto Nakashima, Kaoru Uchimaru

**Affiliations:** Laboratory of Tumor Cell Biology, Department of Computational Biology and Medical Sciences, Graduate School of Frontier Sciences, University of Tokyo, Tokyo 1088639, Japan

**Keywords:** TNFRSF, TNFSF, CD30 (TNFRSF8), CD30L (TNFSF8), trogocytosis, super-enhancer, lymphomagenesis, HTLV-1, ATL

## Abstract

CD30, a member of the tumor necrosis factor receptor superfamily, plays roles in pro-survival signal induction and cell proliferation in peripheral T-cell lymphoma (PTCL) and adult T-cell leukemia/lymphoma (ATL). Previous studies have identified the functional roles of CD30 in CD30-expressing malignant lymphomas, not only PTCL and ATL, but also Hodgkin lymphoma (HL), anaplastic large cell lymphoma (ALCL), and a portion of diffuse large B-cell lymphoma (DLBCL). CD30 expression is often observed in virus-infected cells such as human T-cell leukemia virus type 1 (HTLV-1). HTLV-1 is capable of immortalizing lymphocytes and producing malignancy. Some ATL cases caused by HTLV-1 infection overexpress CD30. However, the molecular mechanism-based relationship between CD30 expression and HTLV-1 infection or ATL progression is unclear. Recent findings have revealed super-enhancer-mediated overexpression at the CD30 locus, CD30 signaling via trogocytosis, and CD30 signaling-induced lymphomagenesis in vivo. Successful anti-CD30 antibody-drug conjugate (ADC) therapy for HL, ALCL, and PTCL supports the biological significance of CD30 in these lymphomas. In this review, we discuss the roles of CD30 overexpression and its functions during ATL progression.

## 1. Introduction

Members of the tumor necrosis factor receptor superfamily (TNFRSF) and TNF superfamily (TNFSF) play crucial roles in both innate and adaptive immunity. It has been suggested that the divergence of the TNFRSF and TNFSF families paralleled the emergence of the adaptive immune system, at least through *en bloc* duplication [1]. As the cloning of the first prototypic members TNF itself and lymphotoxin α, 19 additional TNFSF members and 29 cognate receptors have been identified [2]. In addition to their roles in many important biological processes (development, organogenesis, immunity, cell death, and survival), TNFRSF and TNFSF members are implicated in various acquired or genetic human diseases, ranging from septic shock to autoimmune disorders, allograft rejection, and cancer.

The group of TNFRSF2, 4, 8, 9, 12, 14, and 18 is located on chromosome 1p36.2–36.3, most of which have cognate ligands in TNFSF subfamilies. The biological functions of the group involve T-cell activation, T-cell homeostasis, and T-cell survival [3]. These observations support a co-evolution perspective of TNFRSF and TNFSF families in the immune system. Malignant lymphoma cells express, depending on their immunophenotype, several TNF receptor and ligand superfamily members. B-cell NHLs frequently express CD27/CD27L (TNFRSF7/TNFSF7), CD30 or CD30L (TNFRSF8 or TNFSF8), CD40 (TNFRSF5), and TNFRs/TNF positive (TNFRSF1 and 2/TNFSF2), but T-cell NHLs show expression of CD30, CD40L (TNFSF5), and TNFRs/TNF [4].

CD30, a member of TNFRSF, is a type I single transmembrane protein consisting of 595 amino acids, whose molecular weight is 105–120 kDa [5]. CD30 was initially found to be strongly expressed on Hodgkin and Reed–Sternberg (H-RS) cells of classical Hodgkin lymphoma (cHL) using an anti-Ki-1 antibody. Ki-1, the first anti-CD30 monoclonal antibody, was raised against the cHL patient-derived cell line L428 and observed to react uniquely with primary and cultured H-RS cells [6]. However, soon after, it was found that CD30 is also strongly expressed in a rare subtype of NHL, called anaplastic large cell lymphoma (ALCL) [7]. Later studies showed the expression of CD30 in other pathological conditions, such as viral infection. CD30 expression is often observed in cells infected with HTLV-1, human immunodeficiency virus type 1 (HIV-1), and Epstein–Barr virus (EBV). However, its expression is limited, and normal cells of lymphoid lineage express CD30 only in some activated lymphocytes; furthermore, their expression level is not as high as that of HL cells and ALCL cells. These observations suggest that CD30 is an activation-associated antigen [8,9]. Normal cells of non-lymphoid lineage express CD30 only in uterine decidual cells.

CD30 is strongly expressed in some malignant lymphomas, and its overexpression characterizes cHL and ALCL. Antibody-drug conjugates (ADC) targeting CD30, such as brentuximab vedotin (BV), have shown striking clinical efficiency in cHL and ALCL. CD30 also overexpresses in a portion of peripheral T-cell lymphoma (PTCL), adult T-cell leukemia/lymphoma (ATL), NK lymphoma, and diffuse large B-cell lymphoma (DLBCL), in which clinical studies of BV therapy have been examined.

ATL is a poor prognosis T-cell malignancy that is caused by human T-cell leukemia virus type I (HTLV-1) infection. ATL develops after about 50 years of clinical latency in 2–5% of HTLV-1 carriers [10]; it is estimated that 5–10 million carriers exit worldwide [11]. ATL is endemic in several regions of the world, in particular south-western Japan, the Caribbean basin, and parts of central Africa. Transformation of HTLV-1–infected T cells in vivo is a multistage process, which reflects the status of HTLV-1 infection, i.e., asymptomatic state, smoldering, chronic, lymphoma, or acute type of ATL [12]. The median survival times for each ATL type in the Japanese nationwide, multicenter, hospital-based study are 1815, 778, 305, and 252 days, respectively [13]. The emergence of malignant cells with polylobulated nuclei, (the typical appearance is termed “flower cells”) characterizes ATL. ATL cells show monoclonal integration of HTLV-1 and mainly express CD4, CADM1, CCR4, CD25, and CD45RO, but usually lack CD7 and frequently downregulate CD3 expression. ATL cells acquire abnormalities in the genome, epigenetics, gene expression, and signal transduction.

We reported that HTLV-1–infected CD30-expressing cells increase during the progression of ATL. In this review, we discuss the roles of CD30 with biological significance and the relationship between CD30 with disease progression of ATL.

## 2. The Functions of CD30

### 2.1. CD30L: TNFSF8

The ligand for CD30 (CD30L), also known as CD153, a member of the tumor necrosis factor (TNF) superfamily, is a type II single transmembrane protein consisting of 234 amino acids, whose molecular weight is 26–40 kDa. CD30L was identified and cloned by Smith CA et al. using a soluble fusion protein consisting of the extracellular domain of human CD30 linked to the hinge, C_H_2, and C_H_3 domains of human immunoglobulin G1 heavy chain [14]. CD30L is expressed relatively broadly, including in granulocytes, macrophages, mast cells, and activated lymphocytes. In terms of pathological conditions, this protein is expressed in a subset of myeloid and lymphoid leukemias and in Burkitt lymphoma [9,15].

### 2.2. CD30 Signal Transduction

Stimulation of CD30-expressing cells by CD30L elicits various cellular signaling responses, including proliferation, survival, cytokine secretion, and cell death, depending on the cell type and the cellular differentiation state. These signals are triggered after CD30 binds to the trimeric CD30L expressed on the surface of surrounding cells. The ligation of CD30L to CD30 triggers the recruitment of intracellular adaptor proteins, such as TNFR-associated factor (TRAF) proteins, to the TRAF binding domain of the cytoplasmic tail of trimerized CD30, resulting in further modifications to downstream signaling molecules [9]. TRAF family members are critical signal transducers that relay signals between stimulus-sensing surface receptors and transcription regulators, ultimately altering gene expression [16]. Members of the TRAF family of proteins (TRAF1–6) have been initially identified as modulators of signaling cascades downstream of TNFRSF members through their adaptor function and/or E3 ubiquitin ligase activity [17]. TRAF1, 2, 3, and 5 bind to the cytoplasmic tail of CD30, and TRAFs induce the activation of the transcription factors of the NF-κB family (Figure 1A). NF-κB can be activated via two major pathways: the canonical and non-canonical signaling pathways. CD30 signaling induces the activation of both of these pathways [18]. The canonical NF-κB pathway is regulated by TAK1 kinase activation, which induces the ubiquitination and proteasomal degradation of IκB family members, resulting in the release and nuclear translocation of NF-κB1/p50–RelA/p65 and NF-κB1/p50–c-Rel dimers. On the other hand, the non-canonical NF-κB pathway depends on NF-κB-inducing kinase (NIK) activation. NIK can phosphorylate and activate IKKα, which, in turn, promotes p100 processing to generate NF-κB2/p52 and allows its nuclear translocation with RelB. It has been reported that the overexpression of CD30 induces CD30 signaling independent of CD30L [19].

### 2.3. CD30 Signal Transduction via Trogocytosis

Trogocytosis is a biological process whereby a cell (receiving cell) nibbles membrane fragments from another cell (donor cell), leading to the transfer of cell surface molecules along with membrane fragments. This phenomenon was first observed by Cone et al. over 50 years ago [20]. They noted the presence of allogeneic MHC class II (MHCII) molecules on adoptively transferred T cells. As this seminal observation, numerous researchers have shown that the T cell receptor (TCR) rapidly acquires MHC molecules from antigen-presenting cells (APCs) via the immunological synapse formed at the cell–cell contact area [21,22]. Several recent studies reported that trogocytosis of MHC class I (MHCI) and MHCII occurs not only between T cells and APCs, but also between various cell types, including APCs–natural killer (NK) cells; tumor cells–T; or NK cells; etc. [23,24], suggesting that the type of cell receiving such MHC may impact antigen-specific T cell activation.

Although CD30 ligated by CD30L was thought to work as a signal initiator that remains on the cell surface [25,26], we observed that the process of CD30 signaling is, in fact, a more dynamic phenomenon, similar to the process whereby the TCR acquires MHCs. On the contact surface of CD30L and CD30-expressing cells, CD30L and CD30 form huge clusters, with CD30 extracting CD30L from the adjoining cell, along with part of their plasma membrane, triggering the internalization of the clustered complex. Furthermore, CD30 signaling generates signalosomes, resulting in intracellular signaling [27]. Thus, we concluded that CD30 signals are a trogocytosis-mediated signaling event (Figure 1B).

## 3. CD30 Gene Induction

### 3.1. CD30 Promoter and Transcriptional Factors for CD30 Gene Induction

Human and murine CD30 promoters share a number of common consensus transcription factor binding motifs that may be involved in the regulation of gene expression. EJ Croager et al. revealed the presence of conserved Sp1 and initiator elements [8], both of which have been implicated in positioning transcriptional machinery in genes that lack a TATA box. Further studies revealed that the CD30 gene promoter region is located between −298~+195 bp from the transcription start site (TSS) [28,29].

DNA methylation is one of several epigenetic mechanisms that involve the transfer of a methyl group onto the C5 position of the cytosine to form 5-methylcytosine. CpG islands, clusters of CpG dinucleotides in GC-rich regions, are regions with a high concentration of cytosine–guanine pairs and are found in many gene promoters. DNA methylation controls gene expression by recruiting proteins involved in gene repression or inhibiting the binding of transcription factors to DNA [30]. The CpG island, encompassing 60 CpG dinucleotides, is located in the promoter region, exon 1, and intron 1 of the CD30 gene locus [31]. The CpG island maintains a hypo-methylated state only in lymphoid lineage cells and a hyper-methylated state in other lineage cells. These results suggest that the 60 CpG state on the CD30 promoter, exon 1, and a part of intron 1 controls the transcriptional induction of the CD30 gene (Figure 2A).

Some studies have reported significant transcriptional factors for CD30 gene induction. The AP-1 transcriptional factor JunB binds at −377 to −371 bp from the TSS in the microsatellite sequences and induces transcriptional activity in HL and ALCL [29,32]. Interferon regulatory factor-4 (IRF4) binds to the CD30 promoter region and induces transcriptional activity [33]. Furthermore, this study demonstrated a novel positive feedback loop involving CD30, NF-κB, and IRF4 in PTCL cell lines. The activation of STAT3, the signal transducer and activator of transcription 3, directly regulates CD30 expression by binding the CD30 promoter region in ALCL cell lines [34]. They also showed that STAT3 binds to two highly conserved STAT3 binding sites located approximately 19 kb downstream from the TSS in intron 1, suggesting that they are candidate enhancer regions. Basic Leucine Zipper ATF-like Transcription Factor 3 (BATF3) belongs to the AP-1 transcription factor family along with JunB. BATF3 was recruited to CD30 regulatory regions, as revealed by ChIP and genome-wide ChIP-seq, and BATF3 knockout decreased CD30 expression in ALCL cell lines. These results indicate that BATF3 is one of the essential factors for CD30 induction in ALCL [35].

### 3.2. Super-Enhancer on CD30 Gene Locus

Together with DNA methylation, histone modifications such as acetylation, methylation, phosphorylation, and ubiquitination represent classical epigenetic mechanisms. Generally, transcription start sites of actively transcribed genes are marked by trimethylated H3K4 (H3K4me3) and acetylated H3K27 (H3K27ac), while di- and tri-methylated H3K9 (H3K9me2/3), trimethylated H3K27 (H3K27me3), and trimethylated H4K20 (H4K20me3) are considered gene silencers. H3K27ac is well recognized as a primary epigenetic mark of active enhancers and is required for enhancers to activate the transcription of target genes. Super-enhancers (SEs) are large clusters of enhancers that drive cell-type-specific expression programs that define cellular identity [36]. Thus, SEs are defined by vast clusters of H3K27ac accumulation. SEs play an essential role in cell growth and differentiation not only in normal cells but also in tumor cells. In particular, cancer-specific SEs have been proven to be critical oncogenic driver types of tumor cells [37].

Wong RWJ et al. showed enhancer profiling using primary leukemic samples from ATL, which is a genetically heterogeneous intractable cancer. Analysis of SE-associated genes showed the CD30 gene locus in an SE in both ATL patient samples and an ATL cell line [38]. These results suggest that CD30 gene expression is associated with the characteristics of ATL cells.

HC Liang et al. identified SE loci in ALCL by genome-wide H3K27ac ChIP-seq analysis [35]. The identified SEs were associated with transcription factors and genes considered key pathogenic factors in ALCL, including CD30, IRF4, JUNB, STAT1, and STAT3 in ALCL cell lines. They found that the SE pattern observed in ALCL cell lines was reflected in ALCL primary patient samples, and the CD30 gene locus formed an SE in both ALCL cell lines and ALCL primary patient samples. Taken together, these results supported the significance of CD30 expression and functions in ALCL and ATL (Figure 2B).

## 4. HTLV-1

### 4.1. HTLV-1 Biology

HTLV-1 is a human retrovirus from the genus *Deltaretrovirus* and contains an approximately 9 kb single-strand RNA genome. The retrovirus genome randomly integrates into the host genome. Glucose transporter 1 (GLUT1), neuropilin 1 (NRP1), and heparan sulfate proteoglycans (HSPGs) are reported to be used for HTLV-1 entry, and HTLV-1 is capable of infecting a variety of cell types, primarily CD4^+^ T cells in human carriers; HTLV-1 has also been reported to infect dendritic cells and HSCs. HTLV-1 particles are not detected in the peripheral blood of HTLV-1 carriers, and HTLV-1 infection occurs primarily through cell-to-cell contact, requiring the formation of viral synapses and/or viral biofilm-like structures. The virological synapse is a virus-induced, specialized area of cell-to-cell contact that promotes the directed transmission of the virus between cells. The viral biofilm-like structure is an extracellular matrix component (ECM)-rich structure with infectious viral particles embedded at the surface of infected cells. It is thought that many infected cells are required to establish HTLV-1 infection because the spread of virus particles does not occur in the body. During HTLV-1 integration into the host genome, the 5′ and 3′ ends of HTLV-1 are duplicated, forming long terminal repeats (LTRs), the promoter regions for transcription factor binding. The proviral genome comprises structural genes; gag, pol, and env, flanked by LTR at both ends. The genome also contains the pX region, which has four partially overlapping open reading frames encoding the proteins p12, p13, p30, Rex, and Tax, which are regulatory or accessory genes. The viral genes are transcribed from the 5′ LTR. HTLV-1 also expresses a minus-strand RNA that can encode a basic leucine zipper factor, HTLV-1 bZIP factor (HBZ), and HZB is the only gene that is encoded in the antisense strand and is transcribed from the 3′ LTR. The HTLV-1 genome has the potential to express multiple products by exploiting various strategies such as frameshifting and alternative mRNA splicing.

### 4.2. Tax and HBZ Functions

Tax is also a modulator of cellular gene expression that is involved in the proliferation of T lymphocytes, mainly through the activation of NF-κB pathways. Tax binds the IKKγ/NEMO molecule in the cytoplasm to influence the activity of the IKKα/IKKβ/IKKγ complex [39,40]. Activated IKKα/IKKβ/IKKγ complexes phosphorylate IκBα and then result in nuclear migration of NF-κB. Tax has transforming activity in rodent fibroblasts and primary human lymphocytes, and Tax transgenic mice develop neoplasia including T-cell lymphoma [41]. These results suggest that Tax is involved in immortalizing and transforming the infected cells. Tax is highly immunogenic, and Tax-expressing cells are eliminated by cytotoxic T-cells (CTLs) that target Tax-expressing cells. HTLV-1–infected T-cells shut off Tax expression to avoid the attack of CTLs, which provides an advantage for cell survival. On the other hand, HTLV-1–infected T-cells reactivate HTLV-1 transcription in hypoxic environments in a hypoxia-inducible factor (HIF)-independent manner. These results suggest that HTLV-1–infected T-cells transiently express Tax protein while hiding from immune cells such as CTLs.

HBZ mRNA is constitutively expressed in HTLV-1–infected T-cells, although the expression level is low. HBZ is considered to play an important role in the oncogenic process driving infected cell proliferation, increasing hTERT transcription and inhibiting apoptosis [42]. Not only HBZ protein but also HBZ mRNA promotes the proliferation of infected cells.

HTLV-1–infected T-cells represent a regulatory T-cell (Treg) phenotype, and HBZ induces transcription of CCR4, TIGIT, and FoxP3, which are the cell surface markers and the master transcriptional factor of Treg, respectively. Thus, it seems that HBZ directs HTLV-1–infected T-cell characteristics. HBZ transgenic mice exhibit the onset of T-cell lymphoma and inflammatory diseases such as dermatitis and alveolitis, similar to HTLV-1 infection-related diseases.

Although Tax has oncogenic properties, ATL cells cannot express Tax for 5′ LTR hypermethylation, the deletion of 5′ LTR, or the nucleotide mutations of the pX region in about half of ATL patients. Tax activates various signaling pathways, and the best-known example is the NF-κB pathway, which is constitutively active in primary ATL cells. Thus, ATL cells appear to retain Tax-mediated activation in the absence of Tax but do not appear to require Tax itself for cell survival.

## 5. Adult T-Cell Leukemia/Lymphoma

### 5.1. Genomic Aberrations in ATL

A large-scale genetic analysis of a cohort of 426 ATL cases clarified the landscape of somatic alterations and characterized unique genetic features in ATL cells [43]. This study revealed the strong enrichment of driver lesions in the TCR/NF-κB signaling pathway, of which the most frequently mutated genes were PLCG1 (36%), PRKCB (33%), CARD11 (24%), VAV1 (18%), and IRF4 (14%). These mutations are suggested to enhance NF-κB activity, consistent with constitutive activation of NF-κB in ATL cells. Furthermore, Kogure et al. recently reported the whole-genome landscape of ATL [44]. They discovered two additional frequent alterations, CIC loss-of-function (33%), which may induce T-cell activation, and Rel-truncations (13%), which induce transcriptional upregulation and gain-of-function proteins. The Rel gene encodes the c-Rel protein of the NF-κB subunit, which has two transcriptional activation domains at the c-terminal side [45].

The longitudinal study showed that characteristic genomic and transcriptomic patterns in ATL or pre-onset individuals are associated with subclonal expansion and switches during the clinical timeline. Multistep mutations in TCR, STAT3, and NOTCH pathway genes establish clone-specific transcriptomic abnormalities and further accelerate their proliferative potential to develop highly malignant clones, leading to disease onset and progression [46]. Other researchers also reported accumulated mutations during ATL progression. They showed that HTLV-1–infected T-cell clones carrying key oncogenic driver mutations could be detected in cases of ATL years before the onset of symptoms [47].

### 5.2. HBZ and Super-Enhancer

Nakagawa et al. showed that BATF3 and IRF4 are master regulators of ATL gene expression and proliferation. HBZ drives BATF3 expression by binding to an SE in the BATF3 locus [48]. SEs in primary ATL cells were identified by Wong RWJ et al., and SEs are enriched at genes involved in the T-cell activation pathway, including IL2RA/CD25, CD30, and FYN, reflecting the origin of leukemia cells. Their analysis identified CCR4 and TIAM2 as critical cancer genes in ATL [38].

## 6. The Relationship between CD30 and ATL Progression

### 6.1. CD30 Expression in ATL Cases

Pathological analysis has shown that some ATL cases overexpress CD30. The transformed large cells observed in lymphoma-type cases are often positive for CD30 but negative for ALK translocation. Takeshita M et al. reported that strongly CD30-expressing ATL was grouped into three types: diffuse CD30^+^ anaplastic large cell lymphoma (group 1); pleomorphic-type lymphoma with diffuse CD30 expression (group 2); and pleomorphic-type lymphoma with positive CD30 expression in large cells, but negative in medium-sized and small cells (group 3) [49]. Groups 1 and 2 frequently presented with extranodal tumors and lymph node enlargement greater than 2 cm in diameter, and rarely with leukemic changes, bone marrow involvement, and hypercalcemia. Patients in group 3 rarely had extranodal tumors but frequently had leukemic changes. Although it showed no significant difference, the overall survival in patients with diffuse CD30^+^ lymphoma was better than that of CD30-negative ATL patients. They indicated that diffuse CD30-positive ATL has unusual clinical and immunohistological findings.

Karube K et al. analyzed 490 T-cell and NK-cell lymphoma patients, including 204 ATL patients [50]. In 319 cases, the phenotypic features of lymphoma cells were successfully determined by flow cytometry, and ATL patients were analyzed in 90 cases. As the majority of lymphoma cells are usually larger than normal lymphocytes, they installed a large lymphoid cell gate using forward and side scatters for preliminary identification, combined pan T-cell markers such as CD2, CD3, CD5, and CD7, and focused on tumor cells. They defined CD marker expression as positive, partly positive, or negative when present on, respectively, ≥70%, 20–70%, or <20% of the pan T-cell positive cells. These results indicated that CD30^+^ and partly positive cases account for 39% of ATL cases.

C Bossard et al. reported the CD30 positivity rate using immunohistochemistry (IHC) in 376 PTCLs, including nine ATL patients [51]. CD30 immunostains were scored using a semiquantitative evaluation of the percentage of CD30^+^ tumor cells on a five-tiered scale (IHC scores: score 0, <5% of CD30^+^ tumor cells; score 1, 5 to 24%; score 2, 25 to 49%; score 3, 50 to 75%; score 4, >75%). This study presented a positivity of 55.5% for total positive cases (scores 1–4) in ATL cases.

We reported CD30 positivity in peripheral blood mononuclear cells (PBMCs) in HTLV-1 carriers according to ATL progression using flow cytometry. HTLV-1 infects CD4^+^ lymphocytes, and these cells express CD25. The CD30^+^CD25^+^CD4^+^ subpopulation within peripheral blood, calculated as a percentage of CD4 positive cells, increased according to the ATL progression. Almost all of the CD30^+^ cells were CADM1 positive and polylobulated cells [52].

These results indicated that whether an ATL case was CD30 positive or negative depended on the cut-off value of the CD30 positivity.

### 6.2. The Biological Significance of CD30 in ATL

Higuchi M et al. reported the molecules responsible for the constitutive activation of NF-κB in ATL cells using a retroviral cDNA library from an ATL cell line and a reporter cell line that was easily distinguishable as a positive clone once NF-κB was activated [53]. They obtained several cDNA clones, including full-length CD30, and concluded that the elevated expression of CD30 was one of the factors responsible for constitutive NF-κB activation in ATL cells.

Our study revealed that the CD30^+^ HTLV-1–infected cell population expanded in accordance with disease progression. CD30 signaling induces cell proliferation, polylobulation of nuclei, and abnormal cell division in HTLV-1–infected cell lines, suggesting a contribution to ATL progression [52]. Furthermore, we aimed to elucidate the mechanism of the tumorigenic process involving genomic aberrations in HTLV-1–infected T-cells. We examined the relationship between CD30 signaling and genomic instability using genome-wide comprehensive genome hybridization (CGH) and whole exome sequencing [54]. CD30 signaling induced an increase in intracellular reactive oxygen species (ROS) without inducing apoptosis and promoted DNA double-strand breaks (DSBs) in ATL cell lines and CD30^+^ ATL cells in a CD30 expression level-dependent manner. We demonstrated that CD30 signals induced chromosomal instability by promoting DSBs via increased ROS. Primary CD30^+^ ATL cells stimulated by CD30 ligand acquired focal copy number gains or losses frequently detected in ATL such as IRF4 gain (6p25.3), PAK2 gain (3q29), or PRDM1 loss (6q21). These results strongly highlighted that endogenous signaling by CD30 is an oncogenic signal that promotes ATL progression (Figure 3).

### 6.3. Serum-Soluble CD30 in ATL

An 88-kd soluble form of CD30 (sCD30) has been shown to be released by CD30^+^ cells in vitro and in vivo following the proteolytic cleavage of the corresponding membrane-bound molecule. Increased serum levels of sCD30 have been found in patients with CD30^+^ lymphoid neoplasia, including HD, ALCL, angioimmunoblastic T-cell lymphoma, ATL, and during infectious mononucleosis [55]. Takemoto et al. showed that the serum level of sCD30 is associated with the development of ATL. They considered that soluble CD30 (sCD30) might be another useful marker for the activity and aggressiveness of ATL [56].

## 7. Discussion

Genomic mutation of the CD30 gene occurs by genomic amplification, not point mutation, in ALCL [57]. On the other hand, primary ALCL and a portion of ATL cells form an SE on the CD30 gene locus. These results suggest that CD30 overexpression is necessary for the promotion of lymphoma progression in ALCL and ATL.

Chronic CD30 signaling induces lymphomagenesis in B-cells in an in vivo mouse model [58]. Our results, in which CD30 signaling induced chromosomal instability in a CD30 expression level-dependent manner, supported the importance of CD30 overexpression for lymphoma progression. CD30 is a target for the treatment of HL and ALCL and a promising target for the treatment of CD30-overexpressing lymphomas, including a portion of ATL.

## 8. Conclusions

CD30 overexpression induces CD30 signal amplification and takes advantage of tumor-cell growth in PTCL and ATL cells. Hypo-methylation on the CpG island of the CD30 promoter; transcriptional factors such as JunB, IRF4, Stat3, and BATF3; and CD30-SE are requirements for CD30 overexpression. CD30 signaling amplificated by CD30 overexpression could induce chromosomal instability in CD30^+^ malignant lymphoma cells, as shown in CD30^+^ ATL cells.

## 9. Future Direction

The CD30 overexpression mechanism induced by CD30-SE and the formation of CD30-SE during lymphoma progression remain unclear. Further studies on this mechanism are needed for the development of novel therapies for CD30^+^ lymphomas.

## Figures and Tables

**Figure 1 ijms-24-08731-f001:**
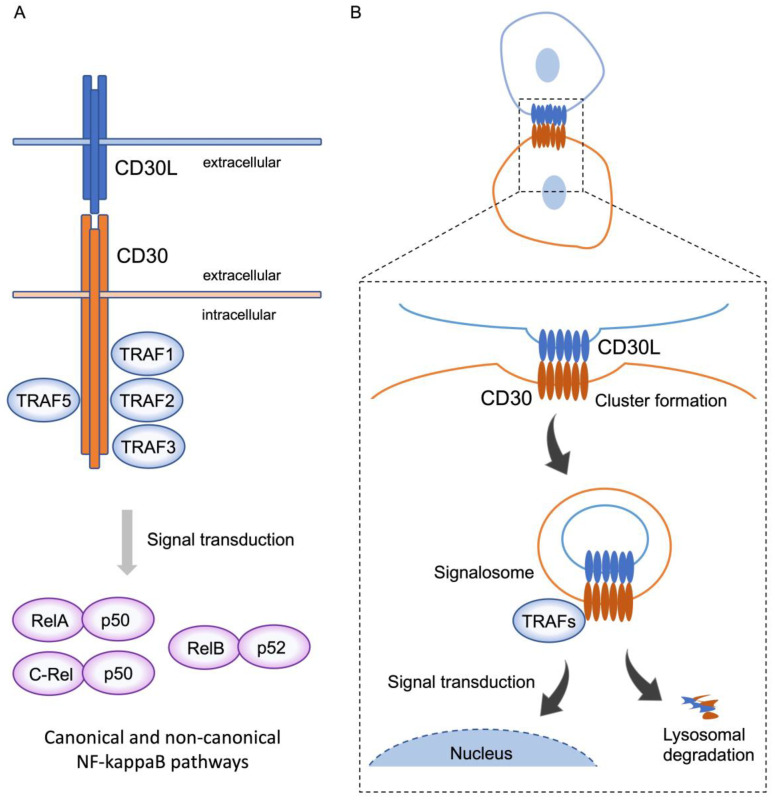
**CD30 signaling via trogocytosis.** (**A**) CD30 binds to trimerized CD30L, and, subsequently, recruits TRAF1, 2, 3, and 5 to the intracellular domain of CD30 molecules. Activation of TRAFs triggers downstream signal transduction and induces nuclear translocation of RelA-p50 and C-rel-p50, and RelB-p52, known as the canonical and non-canonical pathways, respectively. This signal transduction elicits various cellular signaling responses, including proliferation, survival, cytokine secretion, and cell death, depending on the cell type and the cellular differentiation state. (**B**) On the contact surface of CD30L and CD30-expressing cells, CD30L and CD30 form huge clusters, with CD30 extracting CD30L from the adjoining cell, along with part of their plasma membrane, triggering internalization of the clustered complex. These complexes simultaneously generate signalosomes, resulting in intracellular signaling, and are, subsequently, degraded in lysosomes. This phenomenon represents trogocytosis-mediated signal transduction.

**Figure 2 ijms-24-08731-f002:**
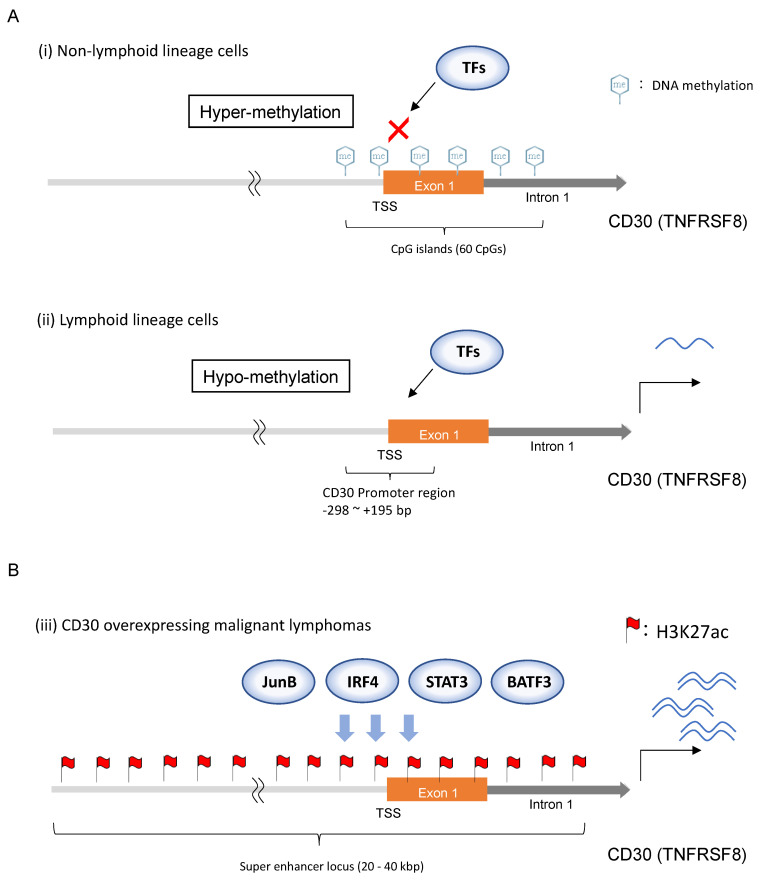
**The mechanisms of CD30 gene induction.** (**A**) The CpG island encompassing 60 CpG dinucleotides is located in the promoter region, exon 1, and intron 1 of the CD30 gene locus. The CpG island maintains a hyper-methylated state in non-lymphoid lineage cells and a hypo-methylated state in lymphoid lineage cells ((**i**) and (**ii**), respectively). These results suggest that the 60 CpG state on the CD30 promoter, exon 1, and a part of intron 1 controls the transcriptional induction of the CD30 gene. (**B**) JunB binds at −377 to −371 bp from the TSS in the microsatellite sequences in HL and ALCL. IRF4, STAT3, and BATF3 bind to the CD30 promoter region in ALCL. These transcriptional factors induce transcriptional activity. Furthermore, the super-enhancer is formed on the CD30 gene locus in ALCL and ATL cell lines and these patient-derived tumor cells.

**Figure 3 ijms-24-08731-f003:**
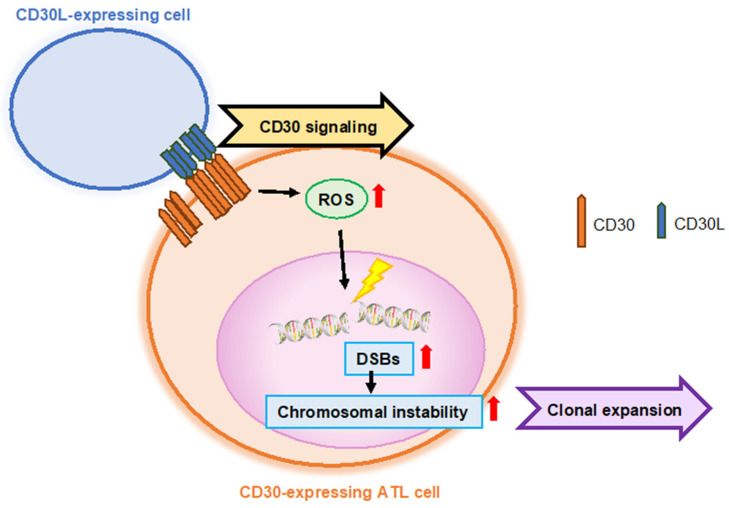
**CD30 signal-induced chromosomal instability in ATL cells.** CD30 signaling triggers chromosomal instability with clonal expansion in the ATL cell line and primary CD30^+^ ATL cells. CD30 signaling induces an increase in intracellular ROS without inducing apoptosis and promotes DSBs in a CD30 expression level-dependent manner. These results suggest that CD30 signaling is one of the oncogenic factors of ATL progression with clonal evolution.

## Data Availability

Not applicable.

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
