# Peer review of "CD30 Expression and Its Functions during the Disease Progression of Adult T-Cell Leukemia/Lymphoma"

_ijms, 2023, doi:10.3390/ijms24108731_

Round 1

Reviewer 1 Report

Dear Editors, 

The title of the review raise the interest of the reader, and clearly sets the scene, so to speak, for a review on the specific lymphoma ATLL, and specifically, on the role of CD30 expression in this lymphoma.  The subject matter is as such of interest and I feel the authors have a high level of expertise in their field.  

However, unfortunately the review lacks focus and the text is not structured in any logical way, making it very difficult to understand the authors overall aim and message.  A review essentially needs to have a clear focus and allow for the reader to build up an understanding, step by step, and end with a summary paragraph that is backed up by the information in the review.

Just to highlight some examples to illustrate the amount of work still required here prior to re-submitting the review: The lymphoma chosen by the authors is not introduced,  nor is there an introduction for T-cell lymphomas in general (even though several T-cell lymphomas are frequently mentioned). The CD30 is also not introduced, rather, the review begins with an introduction to the superfamily to which CD30 belongs which may well be relevant however the text does then not place CD30 within its family, so to speak, and the subsequent paragraphs and sections are not quite ordered in way that can be easily followed. It is as this is a set of very useful notes that can form the basis for creating a draft paper to submit as a review. Indeed, the heading of section 2 reads "relevant sections", suggesting these are initial notes and not a completed draft. Some paragraphs are also seemingly not quite required - just as an example,  section 2.0 - what is the significance of this text, in context of how CD30 biology influences the progression of ATLL?   Hence, my recommendation to the editors is to re-look at a significantly revised manuscript, and I would recommend to the authors to create a draft, and consider to leave out some of the less pertinent paragraphs, arrange the text in a logical order, decide on whether to focus on ATLL or all T-cell lymphomas that may express CD30, and which ever focus is decided, then properly introduce that, then introduce CD30 , and ensure that the test leads up to the conclusion or summary that you intended to highlight with this review. 

Author Response

Response to Reviewer 1 Comments

The title of the review raise the interest of the reader, and clearly sets the scene, so to speak, for a review on the specific lymphoma ATLL, and specifically, on the role of CD30 expression in this lymphoma.  The subject matter is as such of interest and I feel the authors have a high level of expertise in their field.  

However, unfortunately the review lacks focus and the text is not structured in any logical way, making it very difficult to understand the authors overall aim and message.  A review essentially needs to have a clear focus and allow for the reader to build up an understanding, step by step, and end with a summary paragraph that is backed up by the information in the review.

Just to highlight some examples to illustrate the amount of work still required here prior to re-submitting the review: The lymphoma chosen by the authors is not introduced,  nor is there an introduction for T-cell lymphomas in general (even though several T-cell lymphomas are frequently mentioned). The CD30 is also not introduced, rather, the review begins with an introduction to the superfamily to which CD30 belongs which may well be relevant however the text does then not place CD30 within its family, so to speak, and the subsequent paragraphs and sections are not quite ordered in way that can be easily followed. It is as this is a set of very useful notes that can form the basis for creating a draft paper to submit as a review. Indeed, the heading of section 2 reads "relevant sections", suggesting these are initial notes and not a completed draft. Some paragraphs are also seemingly not quite required - just as an example,  section 2.0 - what is the significance of this text, in context of how CD30 biology influences the progression of ATLL?   Hence, my recommendation to the editors is to re-look at a significantly revised manuscript, and I would recommend to the authors to create a draft, and consider to leave out some of the less pertinent paragraphs, arrange the text in a logical order, decide on whether to focus on ATLL or all T-cell lymphomas that may express CD30, and which ever focus is decided, then properly introduce that, then introduce CD30 , and ensure that the test leads up to the conclusion or summary that you intended to highlight with this review. 

Response: In response to the reviewer's comments, we modified the introduction to focus on CD30 and ATL and then relevant sections. The modified structure is as follows.

  1. Introduction (including the paragraphs for CD30 and ATL)
  2. The functions of CD30

2.1: TNFSF8

2.2: CD30 signal transduction,

2.3: CD30 signal transduction via trogocytosis

  1. CD30 gene induction

3.1: CD30 promoter and transcriptional factors for CD30 gene induction,

3.2: Super-enhancer on CD30 gene locus

  1. HTLV-1

4.1: HTLV-1 biology

4.2: Tax and HBZ functions

  1. Adult T-cell leukemia/lymphoma

5.1: Genomic aberrations in ATL

5.2: HBZ and super-enhancer

  1. The relationship between CD30 and ATL progression

6.1: CD30 expression in ATL cases

6.2: The biological significance of CD30 in ATL

6.3: Serum-soluble CD30 in ATL

  1. Discussion
  2. Conclusions
  3. Future Direction

We added the introduction for CD30 and Adult T-cell leukemia/lymphoma (page 2, line 46-60, line 67-79, respectively).

In response to the reviewer's comments, we rewrote the conclusion (page 11, line 391-396). “CD30 overexpression induces CD30 signal amplification and takes advantage of tumor-cell growth in PTCL and ATL cells. Hypo-methylation on the CpG island of CD30 promoter, transcriptional factors such as JunB, IRF4, Stat3, and BATF3, and CD30-SE are requirements for CD30 overexpression. CD30 signaling amplificated by CD30 overexpression could induce chromosomal instability in CD30+ malignant lymphoma cells as shown in CD30+ ATL cells.”

Reviewer 2 Report

The authors conducted an in-depth study to elucidate the role of CD30 in the progression of T-cell lymphoma. Nonetheless, prior to its publication, I noted only a number of minor comments.

1. I suggest revising the abstract to emphasize the role of CD30 in T-cell lymphoma, instead of beginning with a discussion of previous studies.

2.Please include a paragraph for the ATL.

3.Please rephrase this sentence in line 73-75.''However, is .......low''

4.The quality of Figure 1 does not meet the journal's standards. It would be advisable to either replace it with a higher-quality image or generate a new one.

5.Few references are missing ''line 200, the authors mentioned about Wong RWJ et al showed .............

6.Please rewrite the conclusion, it doesn't look conclusive.

Author Response

Response to Reviewer 2 Comments

The authors conducted an in-depth study to elucidate the role of CD30 in the progression of T-cell lymphoma. Nonetheless, prior to its publication, I noted only a number of minor comments.

  1. I suggest revising the abstract to emphasize the role of CD30 in T-cell lymphoma, instead of beginning with a discussion of previous studies.

Response: In response to the reviewer's comments, we modified the abstract to emphasize the role of CD30 in T-cell lymphoma (page 1, line 9-11). “CD30, a member of the tumor necrosis factor receptor superfamily, plays roles in pro-survival signal induction and cell proliferation in peripheral T-cell lymphoma (PTCL) and adult T-cell leukemia/lymphoma (ATL).”

  1. Please include a paragraph for the ATL.

Response: We included the paragraph for the ATL in the introduction and the section for the ATL (page 2, line 67-80, page 8, line 278-306).

  1. Please rephrase this sentence in line 73-75.''However, is .......low''

Response: In response to the reviewer's comments, we rephrase the sentence in line 56-58 in the revised version. “However, its expression is limited, and normal cells of lymphoid lineage express CD30 only in some activated lymphocytes; furthermore, their expression level is not as high as that of HL cells and ALCL cells.”

  1. The quality of Figure 1 does not meet the journal's standards. It would be advisable to either replace it with a higher-quality image or generate a new one.

Response: We generated a new Figure 1 with a higher-quality image in the revised version.

  1. Few references are missing ''line 200, the authors mentioned about Wong RWJ et al showed .............

Response: We added the reference number for Wong RWJ et al. (page 7, line 208-211, page 9, line 303-306).

  1. Please rewrite the conclusion, it doesn't look conclusive.

Response: In response to the reviewer's comments, we rewrote the conclusion (page 11, line 391-396). “CD30 overexpression induces CD30 signal amplification and takes advantage of tumor-cell growth in PTCL and ATL cells. Hypo-methylation on the CpG island of CD30 promoter, transcriptional factors such as JunB, IRF4, Stat3, and BATF3, and CD30-SE are requirements for CD30 overexpression. CD30 signaling amplificated by CD30 overexpression could induce chromosomal instability in CD30+ malignant lymphoma cells as shown in CD30+ ATL cells.”

Round 2

Reviewer 1 Report

Dear Authors and Editors,   I'm afraid I feel that this review still needs extensive editing if it is to become a review that is of interest to a reader who would like to understand the role of CD30 in ATLL.    Since the review is at a stage of requiring editing  I feel it goes beyond the role of a reviewer and enters the area of co-authorship/professional (and paid for) editing.  I of course do not suggest co-authorship or that I should be paid!  I just want to illustrate and highlight that the time available to the average reviewer to carry out actual manuscript peer review (fact checks, relevance of findings,  overall comment on the scientific content including whether relevant references are included) is nil since the reviewer has to spend a huge amount of time working on basic manuscript writing, getting into shape - which to me is not actually the job of a reviewer.    My advice to the authors at this stage is to try to work with questions in order to home in on what their contribution to the journal's readership's knowledge and understanding should be.  To ' be tough' on what they include in the review.  A shorter text with less yet precise information, and with the significance of the included  facts and information placed in context, usually makes for a powerful and interesting  review. On the other hand, a longer text with wide-spread subject areas that are not clearly 'connected' by the authors leaves the reader wondering 'what is the significance of all these facts?'  So.  My advice to the authors is: For each fact /paragraph, ask : What is the significance of this?      For example: "Why does it matter/What is the significance of the fact that EBV infects T-cells and B-cells?"   Well, we know that ATLL, the lymphoma that is the subject of this review, is linked to HTLV1 infection.   What is the connection between EBV infected lymphomas and HLTV-1 infection and ATLL lymphoma (I am not aware of one but maybe there is) Well, this review is all about CD30 and its role in ATLL, so, perhaps both viruses use CD30 to enter the cells? (not insofar as I am aware).   Is EBV infection linked to CD30 in the progression of ATLL?  And so,  if you can't explain how the fact you stated plays a role in the overall message of the review, then just leave it out.   Try to 'connect the dots'. If you have gone to some length to inform the reader of various variants found in ATLL for example - connect those back to CD30. Are any of those variants related to CD30 in any way - what is the connection there? And so on.    Ultimately it is up to the editors to decide on what type of manuscripts they would like to publish of course, and I appreciate there may be different styles wished for by different journals.  Dear Authors and Editors,   I'm afraid I feel that this review still needs extensive editing if it is to become a review that is of interest to a reader who would like to understand the role of CD30 in ATLL.    Since the review is at a stage of requiring editing  I feel it goes beyond the role of a reviewer and enters the area of co-authorship/professional (and paid for) editing.  I of course do not suggest co-authorship or that I should be paid!  I just want to illustrate and highlight that the time available to the average reviewer to carry out actual manuscript peer review (fact checks, relevance of findings,  overall comment on the scientific content including whether relevant references are included) is nil since the reviewer has to spend a huge amount of time working on basic manuscript writing, getting into shape - which to me is not actually the job of a reviewer.    My advice to the authors at this stage is to try to work with questions in order to home in on what their contribution to the journal's readership's knowledge and understanding should be.  To ' be tough' on what they include in the review.  A shorter text with less yet precise information, and with the significance of the included  facts and information placed in context, usually makes for a powerful and interesting  review. On the other hand, a longer text with wide-spread subject areas that are not clearly 'connected' by the authors leaves the reader wondering 'what is the significance of all these facts?'  So.  My advice to the authors is: For each fact /paragraph, ask : What is the significance of this?      For example: "Why does it matter/What is the significance of the fact that EBV infects T-cells and B-cells?"   Well, we know that ATLL, the lymphoma that is the subject of this review, is linked to HTLV1 infection.   What is the connection between EBV infected lymphomas and HLTV-1 infection and ATLL lymphoma (I am not aware of one but maybe there is) Well, this review is all about CD30 and its role in ATLL, so, perhaps both viruses use CD30 to enter the cells? (not insofar as I am aware).   Is EBV infection linked to CD30 in the progression of ATLL?  And so,  if you can't explain how the fact you stated plays a role in the overall message of the review, then just leave it out.   Try to 'connect the dots'. If you have gone to some length to inform the reader of various variants found in ATLL for example - connect those back to CD30. Are any of those variants related to CD30 in any way - what is the connection there? And so on.    Ultimately it is up to the editors to decide on what type of manuscripts they would like to publish of course, and I appreciate there may be different styles wished for by different journals. 

Author Response

Response to Reviewer 1 Comments

Dear Authors and Editors, I'm afraid I feel that this review still needs extensive editing if it is to become a review that is of interest to a reader who would like to understand the role of CD30 in ATLL. Since the review is at a stage of requiring editing I feel it goes beyond the role of a reviewer and enters the area of co-authorship/professional (and paid for) editing. I of course do not suggest co-authorship or that I should be paid! I just want to illustrate and highlight that the time available to the average reviewer to carry out actual manuscript peer review (fact checks, relevance of findings, overall comment on the scientific content including whether relevant references are included) is nil since the reviewer has to spend a huge amount of time working on basic manuscript writing, getting into shape - which to me is not actually the job of a reviewer. My advice to the authors at this stage is to try to work with questions in order to home in on what their contribution to the journal's readership's knowledge and understanding should be.  To ' be tough' on what they include in the review.  A shorter text with less yet precise information, and with the significance of the included facts and information placed in context, usually makes for a powerful and interesting review. On the other hand, a longer text with wide-spread subject areas that are not clearly 'connected' by the authors leaves the reader wondering 'what is the significance of all these facts?' 

So. My advice to the authors is: For each fact /paragraph, ask: What is the significance of this? For example: "Why does it matter/What is the significance of the fact that EBV infects T-cells and B-cells?"   Well, we know that ATLL, the lymphoma that is the subject of this review, is linked to HTLV1 infection. What is the connection between EBV infected lymphomas and HLTV-1 infection and ATLL lymphoma (I am not aware of one but maybe there is) Well, this review is all about CD30 and its role in ATLL, so, perhaps both viruses use CD30 to enter the cells? (not insofar as I am aware). Is EBV infection linked to CD30 in the progression of ATLL?  And so, if you can't explain how the fact you stated plays a role in the overall message of the review, then just leave it out.

Response: Because this review submitted the special issue of “Host and Human Oncovirus Interaction” in IJMS, we included “EBV” in the abstract. As noted in the abstract, CD30 expression is often observed in EBV-infected cells, and EBV-gene proteins such as LMP-1 and EBNA2 upregulate CD30 [1]. Hodgkin-like cells observed in group 3 in section 6.1 are often infected with EBV, and the overall survival in group 3 is worse than that in groups 1 or 2. However, in response to the reviewer’s comments, we removed the sentences related to EBV because EBV is slightly outside the scope of this review in relation to CD30 and ATL.

As modified abstract is below:

“CD30, a member of the tumor necrosis factor receptor superfamily, plays roles in pro-survival signal induction and cell proliferation in peripheral T-cell lymphoma (PTCL) and adult T-cell leukemia/lymphoma (ATL). Previous studies have identified the functional roles of CD30 in CD30-expressing malignant lymphomas, not only PTCL and ATL, but also Hodgkin lymphoma (HL), anaplastic large cell lymphoma (ALCL), and a portion of diffuse large B-cell lymphoma (DLBCL). CD30 expression is often observed in virus-infected cells such as human T-cell leukemia virus type 1 (HTLV-1). HTLV-1 is capable of immortalizing lymphocytes and producing malignancy. Some ATL cases caused by HTLV-1 infection overexpress CD30. However, the molecular mechanism-based relationship between CD30 expression and HTLV-1 infection or ATL progression is unclear. Recent findings have revealed super-enhancer-mediated overexpression at the CD30 locus, CD30 signaling via trogocytosis, and CD30 signaling-induced lymphomagenesis in vivo. Successful anti-CD30 antibody-drug conjugate (ADC) therapy for HL, ALCL, and PTCL supports the biological significance of CD30 in these lymphomas. In this review, we discuss the roles of CD30 overexpression and its functions during ATL progression.”

Try to 'connect the dots'. If you have gone to some length to inform the reader of various variants found in ATLL for example - connect those back to CD30. Are any of those variants related to CD30 in any way - what is the connection there? And so on. 

Response: We previously reported that CD30 signaling induced chromosomal instability, which is associated with the tumorigenic process. In fact, we demonstrated that primary CD30+ ATL cells stimulated by CD30 ligand acquired focal copy number gains or losses frequently detected in ATL such as IRF4 gain (6p25.3), PAK2 gain (3q29), or PRDM1 loss (6q21). We suppose that CD30-induced these variants are involved in ATL progression. In response to the reviewer’s comments, the sentence below was included in section 6.2 in the second revised version (page 10, line 363-365).

“Primary CD30+ ATL cells stimulated by CD30 ligand acquired focal copy number gains or losses frequently detected in ATL such as IRF4 gain (6p25.3), PAK2 gain (3q29), or PRDM1 loss (6q21).”

Ultimately it is up to the editors to decide on what type of manuscripts they would like to publish of course, and I appreciate there may be different styles wished for by different journals. 

  1. Lawrence, J. B.; Villnave, C. A.; Singer, R. H., Sensitive, high-resolution chromatin and chromosome mapping in situ: presence and orientation of two closely integrated copies of EBV in a lymphoma line. Cell 1988, 52, (1), 51-61.